# The Wave Period Parameterization of Ocean Waves and Its Application to Ocean Wave Simulations

Jialei Lv [1], Wenjing Zhang [1,*], Jian Shi [1], Jie Wu [1], Hanshi Wang [1], Xuhui Cao [2], Qianhui Wang [1] and Zeqi Zhao [1]

[1] College of Meteorology and Oceanography, National University of Defense Technology, Changsha 410073, China; lvjialei18@nudt.edu.cn (J.L.); shijian@nudt.edu.cn (J.S.); wujie2022@nudt.edu.cn (J.W.); wanghanshi@nudt.edu.cn (H.W.); wangqianhui21@nudt.edu.cn (Q.W.); zhaozeqi22@nudt.edu.cn (Z.Z.)

[2] School of Hydraulic and Environmental Engineering, Changsha University of Science and Technology, Changsha 410114, China; caoxuhui@stu.csust.edu.cn

\* Correspondence: zhangwenjing21@nudt.edu.cn

**Abstract:** The wave period is a wave parameter that is significantly influenced by factors such as wind speed and bottom topography. Previous research on wave period parameterization has primarily focused on wind-dominated sea areas and may not be applicable to certain regions, such as the equatorial calm or coastal areas dominated by swell waves. To address this limitation, this paper utilizes the third-generation wave numerical model SWAN to perform wave numerical simulations for a portion of the Northwest Pacific Ocean. The simulation incorporates observational data from nearshore stations, buoys, and satellite altimeters for error analysis. To develop a new wave parameterization scheme (WS-23), we employ extensive NDBC buoy data and incorporate the exponential rate and wave age characteristics that were previously established by predecessors. Our scheme introduces a judgement mechanism to distinguish between wind waves, swell waves, and mixed waves. The resulting ocean wave factor enhances the mean wave period values calculated using the model and other parameterization schemes. The experimental results demonstrate that our new parameterization scheme effectively improves the abnormal peak of the fitting data. Comparing the output values of the mean wave period element output of the SWAN model with our new parameterization scheme, we observe a reduction in the mean values of $E_a$, $E_c$, and *RMSE* by 0.231, 1.94%, and 0.162, respectively, while increasing the average $r$ by 0.05.

**Keywords:** numerical simulation; wave model; parameterization; wave periods

## 1. Introduction

Waves are one of the most common phenomena in the ocean. Sea surface waves have a direct influence on all human activities that occur at sea [1]. With the development of computer technology and marine scientific theory, more information is needed about the changing characteristics of wave fields. At present, advanced wave models include the third-generation wave model WAM (Wave Model), SWAN (Simulating Waves Nearshore, commonly used in nearshore areas), and WW3 (WAVEWATCH III, commonly used in the ocean) [2–6].

With the development of wave models, the abundance of assimilation data, and the improvements in grid technology, wave models have been used to solve many scientific problems [7–12]. In wave numerical simulations, many ocean phenomena exist, and it is difficult to obtain results through calculations directly with the accuracy of the grid (not less than 50 m), which is called the sub-grid scale or sub-mesoscale physical process [13–15]. Thus, it is necessary to explore the empirical formulas derived from different physical processes to replace the model and generate efficient data, and the large-scale variable parameterization method used to close the equation is called the parameterization scheme [16,17]. The parametric scheme is an essential part of ocean wave simulation. As the

underlying surface of the atmospheric boundary layer, the sea surface plays an important role in atmosphere–ocean coupling simulations [18,19].

The common parameterization schemes of ocean waves include the parameterization of the wave height, drag coefficient, and wave period. According to previous research results, significant wave height ($H_s$) parameterization is mostly based on the empirical formula by Taylor and Yelland [20], which has been enriched and improved after being optimized through wind speed piecewise fitting and peak wave period ($T_p$) analysis [21–23]. The parameterization of the drag coefficient is widely used. The sea surface drag coefficient is a key factor in numerical simulations under high wind speeds [24], and it is also an important method to obtain other factors via dimensionless analysis [25]. In this study, a parameterization scheme of the drag coefficient based on Wu was formed [26,27] and gradually extended to 100 m/s [28]. The parameterization of the mean wave period ($T_z$) depends on data for function fitting; however, due to the limitation of satellite altimeter and buoy data in three-dimensional space or the time dimension, the fitting effect needs to be improved [29]. First, $T_z$ can be calculated by the correlation between moment zero ($m_0$) and moment four ($m_4$) in the wave spectrum by relying on satellite altimeter data [30,31]. Then, considering the correlation of the sea state and component waves, $T_z$ can be obtained [21,32,33] through buoy observation data, as well as the corresponding relationship between the wave age ($C_p/U_{10}$) and the dimensionless wave height ($H^*$).

Wind waves and swells often appear together in the ocean, and wind energy input is needed in the growth process of wind waves, while swells do not need wind energy input and can react to wind [34]; these differ from each other in terms of energy. In the open ocean, the development of wind waves is only related to the surface wind speed [35]. However, in coastal waters, wave development is affected by topography and local wind speed changes, which make the wave composition complex and unable to be summarized using one scheme. The existing parameterization scheme of the mean wave period is relatively rough, as it only fits the period according to the corresponding relationship between the wave age ($C_p/U_{10}$) and dimensionless wave height ($H^*$), and it only provides one set of parameterization coefficients. The selected buoy fitting data do not distinguish between offshore and nearshore areas, and the types of offshore waves are not fully considered. Based on the characteristics of wave propagation in finite depth sea areas and considering the relationships of energy propagation, the wind wave, swell, and mixed wave are fitted separately, which is also suitable for optimizing the SWAN model. The specific research content of this paper is based on the SWAN wave model through the observation data of three kinds of buoys or stations to test and fit the wave parameter output with the model and obtain a conclusion based on comparisons with previous parameterization schemes.

The introduction of the article is shown in Section 1. Section 2 mainly introduces the data and division of the study sea area. Section 3 details the testing of the model output results and the parametric verification. Section 4 describes the analysis and a summary of the results. Section 5 provides a conclusion for this research.

## 2. Materials and Methods

### 2.1. Data Introduction

The Northwest Pacific Ocean is close to the eastern part of the Asian continent, with a complex coastline and seabed topography and many islands and reefs; it is the birthplace of mesoscale systems and is one of the sea areas with the most frequent occurrence of internal waves. This area is greatly affected by monsoons, and the Northwest Pacific Ocean is also an important typhoon-generating location, with an average of 26 to 40 typhoons generated in this sea area every year, which is the sea area with the most tropical cyclones in the world [36]. Based on the triangle unstructured grid, this paper selects the Northwest Pacific Ocean as the research object, with a longitudinal range of 100° to 140° and a latitudinal range of 5° to 45°. To verify the numerical simulation results of ocean waves, we use typhoon, buoy, and station observation data. The ECMWF (https://cds.climate.copernicus.eu, accessed on 1 July 2023) reanalysis model data are some of the most commonly used

wind field data. ERA5 provides hourly estimates for many atmospheric, ocean wave, and land surface quantities. The temporal resolution of the data is 1 h, the spatial resolution is $0.25° \times 0.25°$, and the file format is ".NC". We analyse the ERA5 interpolated reanalysis wind field to carry out the numerical simulation of ocean waves with the third-generation wave model SWAN as the framework. The detailed information has shown in Figure 1a.

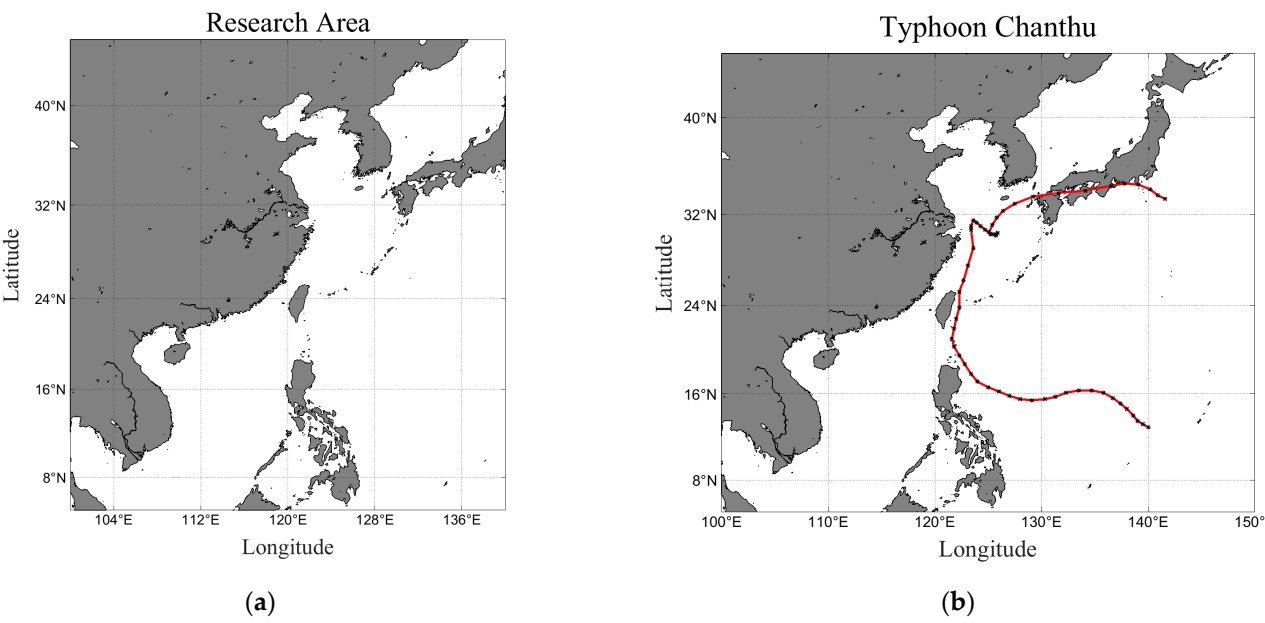

**Figure 1.** (**a**) Research area in the Northwest Pacific Ocean; (**b**) Typhoon Chanthu movement route.

To show better wave simulation effects and adapt the grid range, we select September, when typhoons occur relatively frequently and wave height changes significantly, as well as Chanthu, Typhoon No. 14 in 2021, as the main objects of this numerical simulation shown in Figure 1b. The typhoon data come from the Tropical Cyclone Data Centre of the China Meteorological Administration (https://tcdata.typhoon.org.cn, accessed on 1 July 2023), and the time step of this data set is three hours. It was the strongest typhoon to occur in the Northwest Pacific since the summer of 2021, with a maximum wind speed of 68 m/s near the centre and a central pressure of 905 hPa [37,38]. This typhoon has a long life cycle, complete process, and moderate intensity, and most of its moving areas are in the sea area of this study.

To obtain more significant simulation results, we use the typhoon best track data as a reference and make a comparison with the simulated wave field. The specific typhoon data are shown in Table 1.

The buoy data selected in this paper are from the NDBC (National Data Buoy Centre, https://www.ndbc.noaa.gov/, accessed on 15 July 2023). The NDBC buoy data are not directly obtained via measurement; they are obtained by sensing the dynamic spacetime displacement of the buoy body. The fast Fourier transform (FFT) is applied to the data using the processor [39], and the data are converted from the temporal domain to the frequency domain. After data noise reduction, the nonlinear spectrum can be obtained. Wave parameters, such as the significant wave height and mean wave period, are obtained through derivation and transformation from the wave spectrum.

We select data from 14 shallow-sea buoys (nearshore) with depths greater than 50 m and 12 deep-sea buoys (offshore) with depths greater than 1000 m in 2021, and use the following buoy data to fit the wave parameters. The data that need to be used are the significant wave height, the mean wave period, and the wind speed. Table 2 shows the latitudinal and longitudinal information, as well as the buoy numbers and types in the ocean. In addition, the blue dots in Figure 2 represent deep-sea buoys, and the red dots represent shallow-sea buoys.

**Table 1.** Typhoon Chanthu (2021) best track and other information.

| Time | Level | Latitude | Longitude | Centre Pressure (hPa) | Max Wind Speed (m/s) |
|------|-------|----------|-----------|-----------------------|----------------------|
| 09.06 6:00 | 8 | 14.0 | 138.6 | 1002 | 15 |
| 09.07 6:00 | 10 | 16.1 | 135.8 | 970 | 35 |
| 09.08 6:00 | 15 | 15.7 | 131.3 | 915 | 62 |
| 09.09 6:00 | 18 | 15.8 | 127.0 | 920 | 58 |
| 09.10 6:00 | 16 | 17.8 | 123.4 | 910 | 68 |
| 09.11 6:00 | 18 | 21.0 | 121.6 | 930 | 58 |
| 09.12 6:00 | 16 | 25.2 | 122.3 | 935 | 50 |
| 09.13 6:00 | 15 | 30.7 | 123.4 | 955 | 42 |
| 09.14 6:00 | 12 | 31.0 | 124.3 | 970 | 28 |
| 09.15 6:00 | 10 | 30.4 | 125.9 | 982 | 28 |
| 09.16 6:00 | 11 | 31.1 | 125.4 | 990 | 28 |
| 09.17 6:00 | 10 | 33.5 | 129.2 | 995 | 23 |
| 09.18 6:00 | 8 | 34.6 | 137.8 | 1002 | 15 |
| 09.19 6:00 | 8 | 33.3 | 141.6 | 1008 | 13 |

**Table 2.** For buoy introduction, data were derived from the NDBC (National Data Buoy Centre).

| Name | Longitude | Latitude | Area | Name | Longitude | Latitude | Area |
|------|-----------|----------|------|------|-----------|----------|------|
| 41008 | 80.87°W | 31.40°N | Shallow | 46041 | 124.74°W | 47.35°N | Shallow |
| 41013 | 77.76°W | 33.44°N | Shallow | 41044 | 58.63°W | 21.58°N | Deep |
| 41025 | 75.45°W | 35.01°N | Shallow | 41046 | 68.34°W | 23.82°N | Deep |
| 42012 | 87.55°W | 30.06°N | Shallow | 41048 | 69.57°W | 31.83°N | Deep |
| 42020 | 96.69°W | 26.97°N | Shallow | 41049 | 62.94°W | 27.49°N | Deep |
| 42036 | 84.51°W | 28.50°N | Shallow | 46006 | 137.38°W | 40.76°N | Deep |
| 42040 | 88.24°W | 29.21°N | Shallow | 46035 | 177.03°W | 57.02°N | Deep |
| 44008 | 69.25°W | 40.50°N | Shallow | 46059 | 129.97°W | 38.05°N | Deep |
| 44013 | 70.65°W | 42.35°N | Shallow | 46073 | 172.01°W | 55.01°N | Deep |
| 44018 | 70.15°W | 42.20°N | Shallow | 46078 | 152.64°W | 55.58°N | Deep |
| 44025 | 73.16°W | 40.25°N | Shallow | 51000 | 153.79°W | 23.53°N | Deep |
| 44027 | 67.30°W | 44.28°N | Shallow | 51002 | 157.75°W | 17.04°N | Deep |
| 44066 | 72.64°W | 39.62°N | Shallow | 51003 | 160.64°W | 19.20°N | Deep |

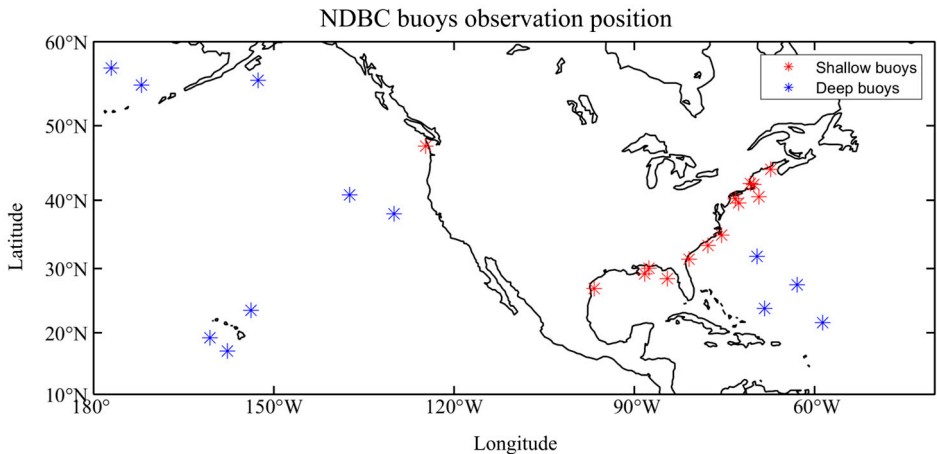

**Figure 2.** NDBC buoy observation positions in the Atlantic Ocean and Pacific Ocean.

We also use Jason-3 altimeter data, which come from the National Centres for Environmental Information (NCEI, https://www.ncei.noaa.gov, accessed on 15 July 2023). The Jason-3 satellite altimeter orbit route has shown in Figure 3. Through orbital measurement data, we use significant wave height ($H_s$) data in the Ku band as another model comparison data. Three lines are selected, including six scanning tracks. They are Cycle205-101, Cycle205-190, Cycle206-203, Cycle206-214, Cycle207-229, and Cycle207-240. The data of

these six sets of altimeters were collected on 09.05, 09.08, 09.19, 09.20, 09.29, and 09.30 in 2021. These six trajectories pass through the studied sea area, and continuous spatial points can be selected from the trajectories as model comparison data. By comparing the altimeter data, we can also test the accuracy of the SWAN model to simulate waves in the deep sea or ocean.

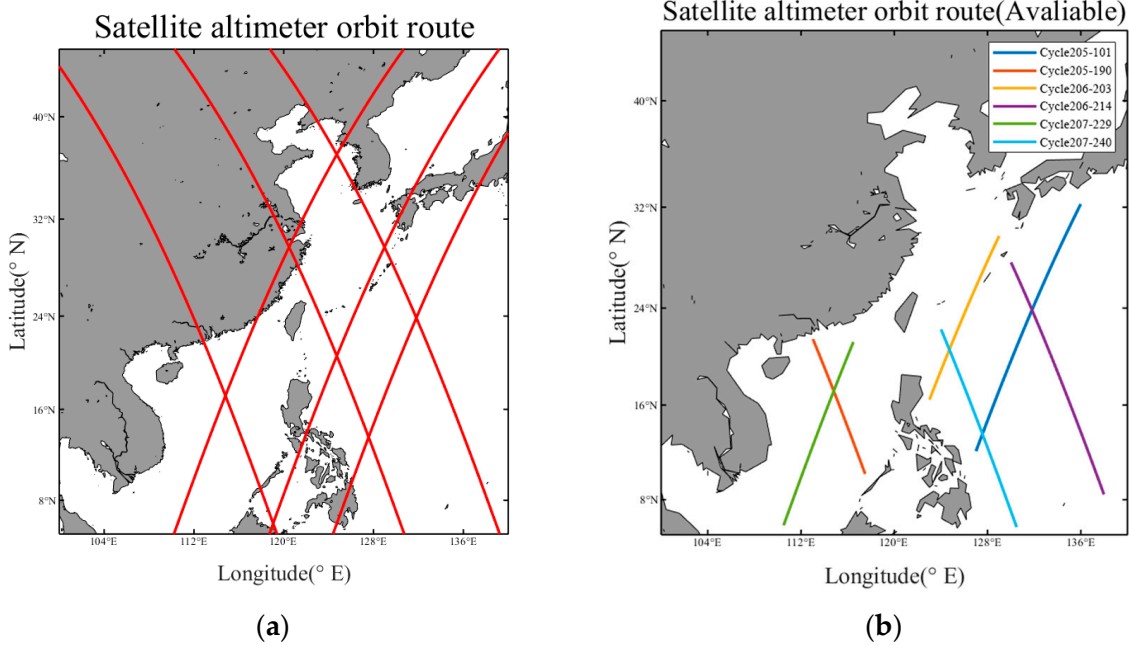

(**a**)　　　　　　　　　　　　　　　　　　　　　　　　　　　(**b**)

**Figure 3.** Jason-3 satellite altimeter orbit route from the NCEI (**a**) and available data (**b**).

To test the optimization degree of the parameterization scheme, we introduce the Korean buoy data, which come from the Korea Meteorological Administration (https://data.kma.go. kr/data/sea, accessed on 15 July 2023), including eight sets of buoy data, all of which are from September 2021. The buoy generates data each hour, and the output parameters include significant wave heights, the mean wave period, wave direction, wind direction, air pressure, temperature, and so on. The buoy data have not been added to the data set of the parameterization scheme, and the data will be used to verify the fitting of the shallow-sea wave parameters. Specific information about the Korea Meteorological Administration buoys is shown in Table 3.

**Table 3.** Korea Meteorological Administration buoy introduction.

| Name | Longitude | Latitude | Element | Time |
|---|---|---|---|---|
| 22101 | 126.01°E | 37.24°N | wind speed, $H_s$ and $T_z$ | 2021.09 |
| 22102 | 125.77°E | 34.79°N | wind speed, $H_s$ and $T_z$ | 2021.09 |
| 22103 | 127.50°E | 34.00°N | wind speed, $H_s$ and $T_z$ | 2021.09 |
| 22105 | 129.95°E | 37.48°N | wind speed, $H_s$ and $T_z$ | 2021.09 |
| 22106 | 129.78°E | 36.25°N | wind speed, $H_s$ and $T_z$ | 2021.09 |
| 22108 | 125.75°E | 36.25°N | wind speed, $H_s$ and $T_z$ | 2021.09 |
| 22188 | 128.23°E | 34.39°N | wind speed, $H_s$ and $T_z$ | 2021.09 |
| 21229 | 131.11°E | 37.46°N | wind speed, $H_s$ and $T_z$ | 2021.09 |

In addition, this paper also selects other data, such as the Global Self-consistent, Hierarchical, High-resolution Geography Database (GSHHG, http://www.ngdc.noaa.gov/ mgg/shorelines/data/gshhs/, accessed on 1 July 2023), unstructured grid data in the northwest Pacific Ocean and six nearshore station observation data in nearshore from National Marine Data Centre (NMDC, https://mds.nmdis.org.cn/, accessed on 1 July 2023), include Station-DCN, Station-LHT, Station-LYG, Station-XCS, Station-XMD, and Station-NJI. All

nearshore stations are in the China Sea. The observation frequency is the same as the KMA buoy and the NDBC buoy. These data are used in the numerical model operation, verification, drawing figures, and other steps in this paper.

### 2.2. SWAN Model Introduction

SWAN (Simulate Waves Nearshore) is the third-generation numerical model of shallow-sea waves developed by Delft University of Technology in the Netherlands (Fluidmechanics Section—(https://swanmodel.sourceforge.io/, accessed on 1 July 2023)). This model has shown great potential in simulating waves nearshore or in shallow water areas that compute random, short-crested, wind-generated waves in coastal regions and inland waters. It is suitable for wave simulation in shallow seas and nearshore areas, including coasts, lakes, and estuaries. It has been continuously updated from its first published version, SWAN 30.51, to the latest version, SWAN 41.45, while improvements are still in progress [40]. SWAN can be nested with the WAVEWATCH III and WAM models, or it can nest grids within itself to drive the simulation.

### 2.2.1. Ocean Wave Energy Spectrum Equation

When considering the influence of the flow field, the spectral energy density is not conserved, but the wave action, $N(\sigma, \theta)$, is conserved. $N(\sigma, \theta)$ varies with time and space. The relationship between $N(\sigma, \theta)$ and $E(\sigma, \theta)$ is $N(\sigma, \theta) = E(\sigma, \theta)/\sigma$. In the Cartesian coordinate system, the wave action equilibrium equation can be expressed as follows:

$$\frac{\partial}{\partial t}N + \frac{\partial}{\partial x}C_x N + \frac{\partial}{\partial y}C_y N + \frac{\partial}{\partial \sigma}C_\sigma N + \frac{\partial}{\partial \theta}C_\theta N = \frac{S}{\sigma} \tag{1}$$

The first term on the left-hand side represents the change rate of the wave action, $N$, with time; the second and third terms are the propagation of the wave action in the $X$ and $Y$ spatial directions, respectively. Moreover, the fourth term is the change in wave action caused by the flow field and water depth in $\sigma$ space; the fifth term gives the propagation of wave action in the $\theta$ direction, which represents the refraction change in water depth due to the flow field. On the right-hand side of the above equation, $S$ is the source and sink term.

This wave model includes the energy input and output terms caused by the wind energy input term, wave–wave interactions, the bottom friction term, the crown dissipation term, and the deep fragmentation term. All the source function terms ($S$) in the SWAN energy equation are given as follows:

$$S = S_{in} + S_{dis} + S_{nl4} + S_{bot} + S_{br} + S_{nl3} + S_{xx} \tag{2}$$

From (3), $S$ is the total wave energy term and $S_{in}$ is the energy input term, which mainly refers to the wind energy input, $S_{in}$. The remaining source function terms are energy dissipation terms. The main dissipative terms are the white crown dissipative term ($S_{dis}$), the bottom friction term ($S_{bot}$), and the wave–wave interaction term ($S_{nl}$). There are two kinds of wave–wave interaction terms, namely the $S_{nl4}$ four-wave interaction term and $S_{nl3}$ three-wave interaction term. Simultaneously, the water depth-induced fragmentation terms ($S_{br}$) and other action terms ($S_{xx}$) must be added. In the deep sea, $S_{dis}$ and $S_{nl4}$ are the source functions leading to wave energy dissipation, and $S_{dis}$ is the main term. In the middle-depth ocean (or in finite depth), the main energy dissipation term is $S_{bot}$. $S_{nl3}$ and $S_{br}$ play a major role in wave energy dissipation in shallow sea or coastal waters, which are also advantages of the SWAN model compared with other wave models. Other wave models do not have the simulation effect of these two source functions.

### 2.2.2. Original Function Term Introduction

From the perspective of energy input and output, (2) can be simplified into (3):

$$S = S_{in}(\sigma, \theta) + S_{ds} + S_{nl} \tag{3}$$

The above equation represents the original function terms. On the right side of the equation, $S_{in}(\sigma, \theta)$ is the wind input term; $S_{ds}$ is the dissipation term, including the energy dissipation caused by white crown dissipation, bottom friction, and depth-induced crushing; and $S_{nl}$ is the wave interaction term.

Currently, there are two different types of computational mechanisms used to determine $S_{in}(\sigma, \theta)$ in wave modelling. The first mechanism is the Phillips resonance mechanism, which represents the linear growth of waves over time. The second mechanism is the Miles instability mechanism, which represents the exponential growth of waves over time. The SWAN model employs a combination of both mechanisms. During the initial stage of wave generation, the Phillips resonance mechanism is used to handle wind energy input. Once the waves have fully developed, the Miles parallel flow instability mechanism is employed to continue processing the wind energy input.

$$S_{in}(\sigma, \theta) = A + BE(\sigma, \theta) \tag{4}$$

In (5), $A$ represents linear growth and $BE(\sigma, \theta)$ represents exponential growth.

The SWAN model considers three types of dissipation effects: the white-capping dissipation term ($S_{dis}$), the bottom friction term ($S_{bot}$), and the depth-induced breaking term ($S_{br}$).

$$S_{ds} = S_{dis} + S_{bot} + S_{br} \tag{5}$$

Under the continuous action of steady wind, the waves continuously grow and eventually reach a fully developed state. At this point, the wave crest undergoes breaking, resulting in whitecaps on the ocean surface. After the whitecaps decay, they take some time to dissipate. The whitecap dissipation term in the SWAN model describes the energy loss due to deep-water wave breaking, with its steepness controlling the degree of dissipation. The formula for the whitecap term in the SWAN model is expressed in terms of the wavenumber, as follows:

$$S_{dis}(\sigma, \theta) = -\Gamma \tilde{\sigma} \frac{\tilde{k}}{k} E(\sigma, d) \tag{6}$$

where $d$ represents the water depth and $\Gamma$ denotes the wave steepness coefficient. The formula is shown as follows. Among them, $C_{ds}$, $\delta$, and $p$ are all adjustable parameters. $\sigma$ and $k$ represent the average wave beam and average frequency, respectively, from which $S_{dis}$ can be obtained as follows:

$$\Gamma = C_{ds}\left( (1 - \delta) + \delta \frac{\tilde{k}}{k} \right) \left( \frac{\tilde{S}}{\tilde{S}_{PM}} \right)^p \tag{7}$$

The SWAN model includes three types of frictional drag models: JONSWAP, Collins, and vortex viscous. The formula for frictional drag can be expressed as follows:

$$S_{bot}(\sigma, \theta) = -C_{bot} \frac{\sigma^2}{g^2 sinh^2(kd)} E(\sigma, \theta) \tag{8}$$

In (8), $\sigma$, $k$, and $\theta$ denote the frequency, wavenumber, and wave direction, respectively. $C_{bot}$ represents for the bottom friction coefficient.

The spectral directional dissipation calculation formula included in the SWAN model is shown as follows, where $D_{tot}$ represents the average dissipation rate of energy per unit area due to wave shallowing and $E_{tot}$ represents the total energy.

$$S_{br}(\sigma, \theta) = -D_{tot} \frac{E(\sigma, \theta)}{E_{tot}} \tag{9}$$

Last, the wave–wave nonlinear interaction refers to the process of energy redistribution between resonance waves with different wave frequencies after wind waves acquire energy.

It is an important physical mechanism for the growth and development of ocean wave processes and can be divided into three-wave interactions and four-wave interactions. In deep water conditions, the four-wave interaction process is more important, while in shallow water conditions, the three-wave interaction process is more significant. The four-wave interaction mainly transfers energy from high frequency to low frequency, while the three-wave interaction mainly transfers energy from low frequency to high frequency.

The SWAN model utilizes a combination of three-wave and four-wave interactions to perform nonlinear calculations of wave interactions. The SWAN model employs Hasselmann's discrete interaction approximation (DIA) scheme to solve the four-wave interaction and Eldeberky's lumped triad approximation (LTA) scheme to solve the three-wave interaction [41].

*2.3. Statistical Analysis Test*

In this paper, we will also use statistical metrics methods to test the accuracy of the simulation data. The statistical variables needed are the average absolute error ($E_a$), average relative error ($E_c$), root mean square error ($RMSE$), and correlation coefficient ($r$). The relevant calculation formulas are given below, where $y_i$ represents the observation data and $x_i$ represents the SWAN simulation and other scheme data. $\overline{x}$, $\overline{y}$ is the mean of the corresponding data.

$$E_a = \frac{1}{N}\sum_{i=1}^{N}|x_i - y_i| \tag{10}$$

$$E_c = \frac{E_a}{\overline{x}} = \frac{\sum_{i=1}^{N}|x_i - y_i|}{\sum_{i=1}^{N}x_i} \tag{11}$$

$$RMSE = \sqrt{\frac{1}{N}\sum_{i=1}^{N}(x_i - y_i)^2} \tag{12}$$

$$r = \frac{\sum_{i=1}^{N}(x_i - \overline{x})(y_i - \overline{y})}{\sqrt{\left[\sum_{i=1}^{N}(x_i - \overline{x})^2\right]\left[\sum_{i=1}^{N}(y_i - \overline{y})^2\right]}} \tag{13}$$

$E_a$ and $E_c$ are used to show the reliability of the measurement results, $RMSE$ is used to describe the dispersion degree of a group of data, and $r$ is a coefficient indicating the linear correlation between different data.

## 3. Results

The results of the model operation and the verification contents are discussed in this section. The first part of this section provides the results and test of the SWAN model, including the simulation analysis comparison with the ERA5 reanalysis wind data, and the wind speed comparison between the SWAN output data and ERA5 reanalysis data. The second and third parts of this section include the verification in nearshore and offshore areas, respectively, with comparisons between observation station data and satellite altimeter data.

*3.1. Validation between Wind Data and SWAN Wave Field Results*

In this section, we verify and analyse the pattern output results. The change in the typhoon wave field is analysed and compared with the wind speed.

Figure 4 shows the wind field output with the model with a significant wave height background field. The red line indicates the typhoon's movement route, and the black solid point indicates the typhoon's location at that time. In the figure, the structure of the typhoon circle formed through numerical simulation is relatively complete, the position of the circle is obvious, the wind field corresponds well with the position of the circle, and the logical centre generated with the typhoon circle forms a good correspondence with the measured centre position.

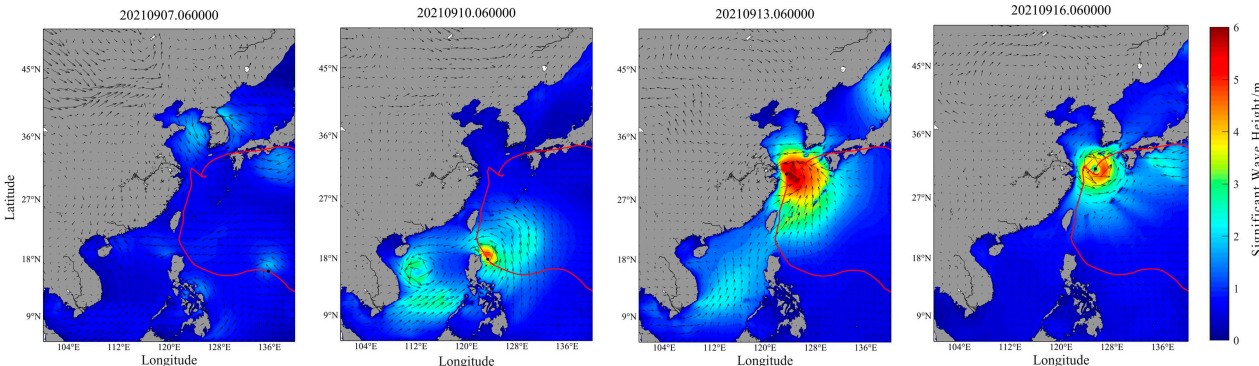

**Figure 4.** Typhoon Chanthu numerical simulation with significant wave height in the Northwest Pacific Ocean.

Then, we compared the wind speed, which is divided into a U component in the horizontal direction and a V component in the vertical direction. In Figure 5, the X coordinate is the wind speed component of the model output, and the Y coordinate is the custom component of the ERA5 reanalysis wind field. To achieve the best test effect in terms of time, we chose one time every three days from 4 September 2021 to 25 September 2021 and made a comparison at each chosen time to distinguish the U direction from the V direction. We calculated the correlation coefficient, and found that the r values of all the data were above 0.97. The minimum r was 0.972, and the maximum r was 0.990, which demonstrates that the SWAN output wind data and reanalysis wind data are highly consistent, and the correlation between the wind field data output with the model and the reanalysis data is strong. It can be concluded from the figure that, compared with the wind field driven with the same model, the wind field output of the model has regression stability.

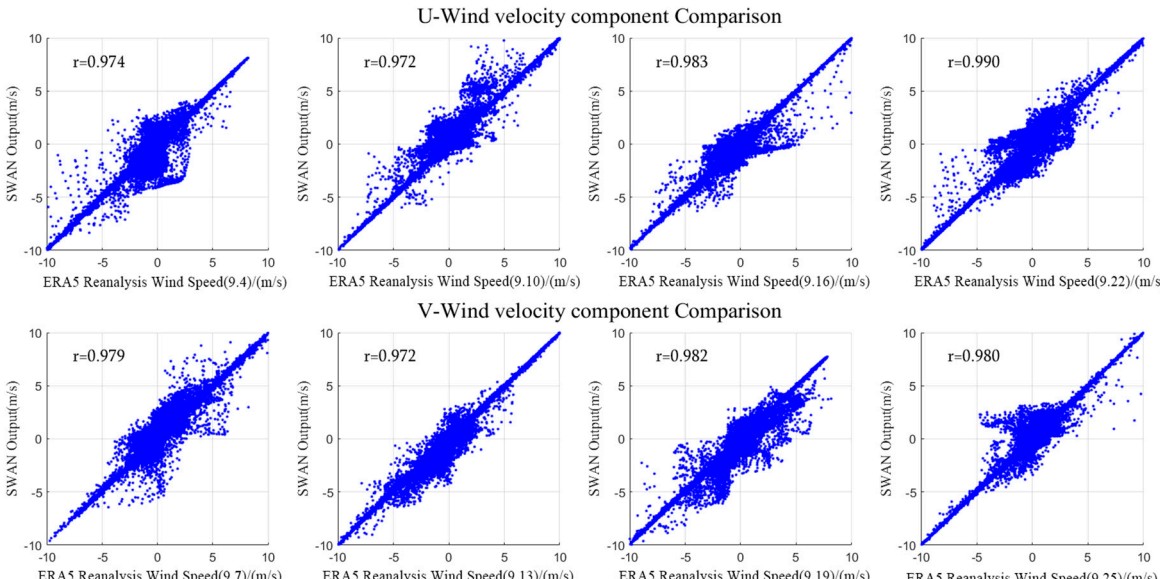

**Figure 5.** Wind speed result from the output of SWAN and ERA5 reanalysis wind data.

### 3.2. Validation with NMDC Station Observation Data

In this part, we use the observation data of nearshore stations to analyse the results of the SWAN output. The meaning is to verify the simulation ability of SWAN in nearshore areas. We use station observation data for nearshore analysis and altimeter data for offshore analysis.

In Figure 6, the data from six observation stations are compared with the model data. Red dots represent the measured values of the station data, and black lines represent the

output of the simulated values of the SWAN model. The $H_s$ parameter simulated using SWAN is in good agreement, and the corresponding trend and numerical value of each factor are obtained. Some stations have poor results, such as the XMD station, and the spectral peak simulation effect is poor. According to Table 4 and Figure 6, we conclude that NJI, LHT, and DCN have the best fitting effect, with $E_a$ values not higher than 0.3 and $E_c$ values not higher than 30%. The $RMSE$ is also within 0.4. The $r$ of all stations is over 0.65. Among them, the best simulation effect is NJI, with an $r$ reaching 0.93.

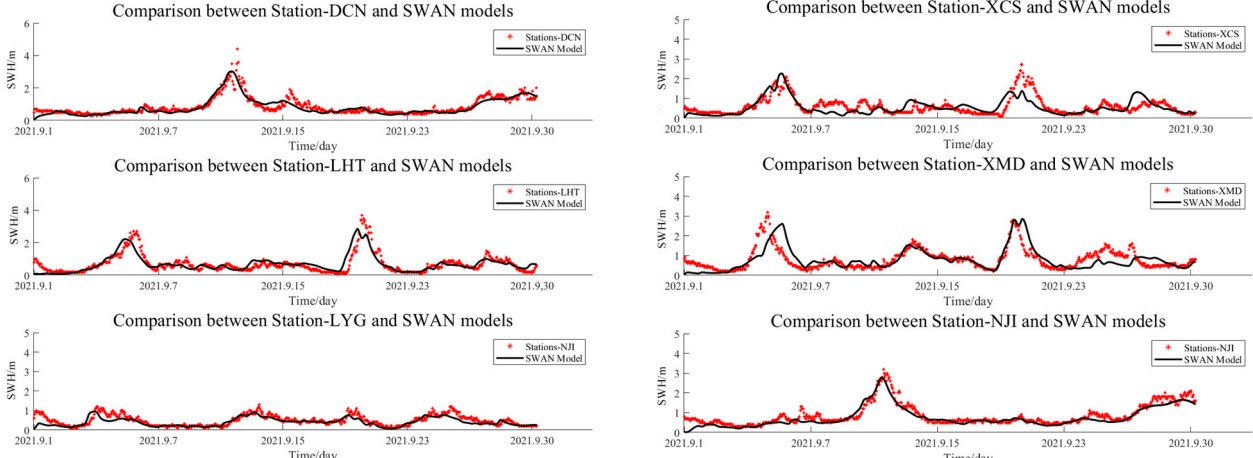

**Figure 6.** Significant wave height results for the output of the SWAN model and station observation data.

**Table 4.** Error analysis between SWAN output and station data.

| Station | $E_a$ | | $E_c$ | | $RMSE$ | | $r$ | |
|---|---|---|---|---|---|---|---|---|
| | $H_s$ | $T_z$ | $H_s$ | $T_z$ | $H_s$ | $T_z$ | $H_s$ | $T_z$ |
| DCN | 0.170 | 2.416 | 0.203 | 0.369 | 0.241 | 2.631 | 0.905 | 0.652 |
| LHT | 0.229 | 1.734 | 0.445 | 0.324 | 0.339 | 2.156 | 0.820 | 0.521 |
| LYG | 0.136 | 1.879 | 0.290 | 0.384 | 0.192 | 2.145 | 0.773 | 0.380 |
| XCS | 0.254 | 1.332 | 0.495 | 0.311 | 0.356 | 1.554 | 0.655 | 0.434 |
| XMD | 0.302 | 1.947 | 0.410 | 0.354 | 0.439 | 2.249 | 0.693 | 0.627 |
| NJI | 0.163 | 2.551 | 0.198 | 0.388 | 0.225 | 2.787 | 0.930 | 0.650 |

In Figure 7, the blue solid point represents the observation data of the station, while the black solid line represents the output value of the SWAN model. Figure 7 shows that the behaviours of the two curves are roughly similar, but the observed value of the station is significantly different from the $T_z$ value of the SWAN output coefficient. The simulated value of the mean wave period of the SWAN output is generally lower than the observed value of the station. The data in Table 4 are the error analysis and comparison between the $H_s$ and $T_z$ output of the model and the station data, to test the accuracy of the model in nearshore simulation. From Table 4, the results of $E_a$, $E_c$, and $RMSE$ show that XCS has a good simulation effect, but the $r$ is only 0.434. All $r$ values are lower than 0.7, which demonstrates that the correlation of the $T_z$ curves is not as strong. All data sets with $E_c$ are lower than 0.4, and all $RMSE$ of the data mostly exceed 2. The maximum $r$ only reaches 0.652, which is the DCN of the stations. Figure and Table prove that the $T_z$ simulated using SWAN has a less than satisfactory simulation effect in coastal waters.

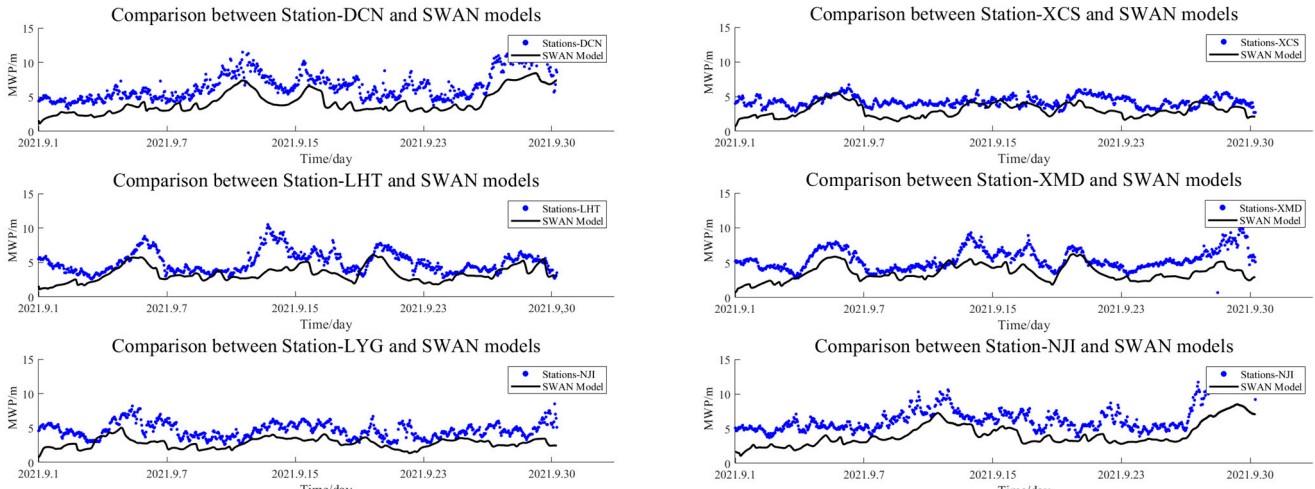

**Figure 7.** Mean wave period results for the output of the SWAN model and station observation data.

From the error analysis table of the model and station data, we can conclude that the simulation ability of SWAN in shallow sea areas needs to be improved, and the simulation ability of $H_s$ is better, while the simulation ability of the mean wave period is worse. From the analysis data of $E_c$ and $RMSE$, the simulation effect of the mean wave period is significantly different from the measured data, and the parametric scheme will be considered later to improve the model output effect.

### 3.3. Validation with NCEI Satellite Data

The offshore comparison data are altimeter data (from NCEI, Jason-3, in Figure 3). The purpose of introducing data is to test the simulation effect of the SWAN model in the ocean. We introduce six satellite altimeter scanning orbits and take orbit measurement data at six different times for verification. We calculated the $E_c$ between the two sets of data to test the simulation ability of the SWAN model.

Figure 8 shows the comparison curve between altimeter data and model data. The blue dot shows the $H_s$ data measured with the altimeter, and the black line segment indicates the $H_s$ data output of the model. The calculated $E_c$ is marked in Figure 8. From the calculated $E_c$, the $E_c$ of all comparison data is less than 25%, and the smallest $E_c$ is Cycle206-214, which is less than 0.12 and reaches 0.1185. The error analysis results show that SWAN has a good simulation effect in the deep sea or ocean. Most of the curve peaks are well simulated, but some curves have poor peak fitting results, such as Cycle206-203 and Cycle207-240. In Figure 8, near the route with poor simulation results, the simulation ability of the wave peak is poor because of the influence of topography that cannot be distinguished by the model and unstructured grid; however, the simulation results are better in most of the South China Sea, northern Philippines, and northern Japan. The best one is Cycle206-214, and the $E_c$ is the smallest, but the crest simulation ability is relatively insufficient.

After comparing the three groups of station data and altimeter data, a typhoon process is selected for trajectory tracking, and the characteristics of the wave field are analysed. Within the allowable error range, the stability test results of the data sets and grids are good, and there is no computational instability. The correlation coefficient ($r$) of the horizontal and vertical components of the wind speed output from the model is kept above 0.95, and the $E_a$ of the altimeter height data is approximately 20%. Compared with the station data, the $H_s$ data simulate well, but the error of the mean wave period is large. Because the mean wave period of SWAN output is a parameter output from the angle of spectral integration, it is not considered. The new parameterization of the mean wave period will be discussed in detail in the Discussion section of this paper.

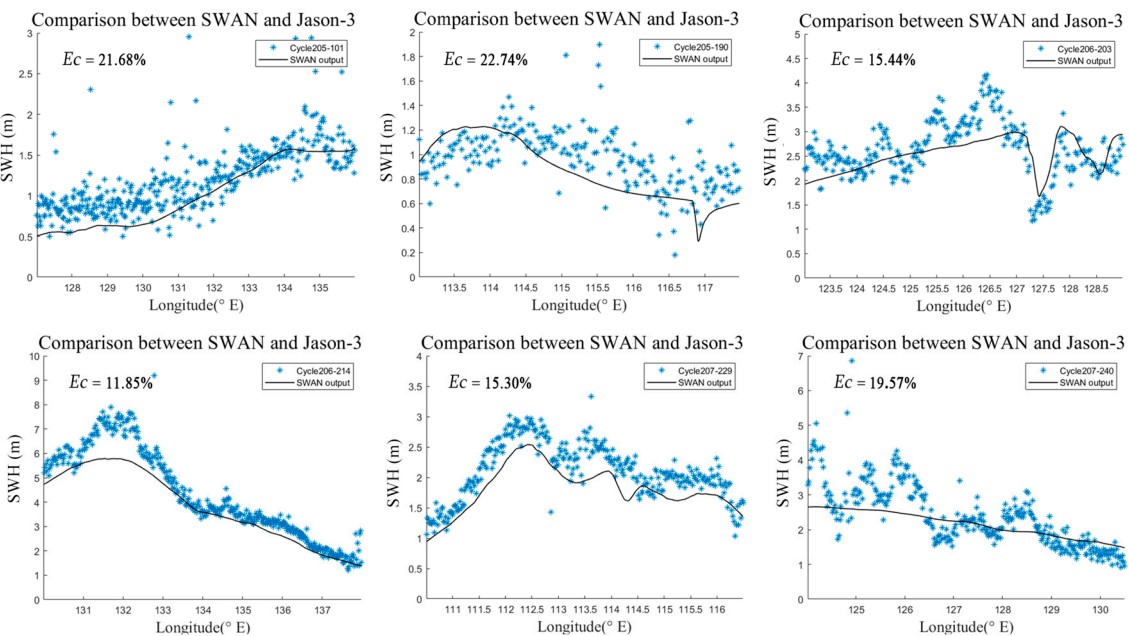

**Figure 8.** Jason-3 significant wave height validation with SWAN output.

## 4. Discussion

In the discussion, the first part of this section reproduces the parameterization of the $T_z$ elements from previous research. Toba's 3/2 exponential rate, Zhao's 3/5 exponential rate, and Wang's new parameters were fitted with the NDBC buoys near Hawaii. In the second part of this section, combined with previous research results, the composition and characteristics of waves are analysed, and the characteristics of wind waves, swells, and mixed waves are distinguished to form a new parameterization scheme, which optimizes the mean wave period output of SWAN.

### 4.1. Previous Parametric Scheme Results

Toba first established a relationship from equation $H^* = BT^{*\frac{3}{2}}$, and proposed a three-second power law for wind waves based on the dimensionless wind wave growth process [42]:

$$\frac{gH}{U^2} = B'\left(\frac{C_p}{U}\right)^{\frac{3}{2}}, \quad B' = (2\pi)^{\frac{3}{2}}\gamma^{\frac{3}{2}}B \tag{14}$$

where $U$ is $u_*$. If an approximate value of $\gamma$ of 0.040 is used, the value of B is determined to be 0.059. Using well-known empirical formulas for equilibrium wind waves, $\frac{gH}{U^2} = 0.30$ and $\frac{C_p}{U} = \frac{gT}{2\pi U} = 1.37$. The value of $B'$ in Equation (15) is determined to be 0.20. Then, a parameter is established in the following equation:

$$\frac{gH}{u_*^2} = 0.20\left(\frac{C_p}{u_*}\right)^{\frac{3}{2}} \tag{15}$$

Zhao believes that satellite altimetry only provides information on wind speed and wave height but does not provide information on wave period and frequency, so they used buoy data from the Japan Meteorological Agency and the least squares method to fit through Equation (16).

$$\frac{C_p}{U_{10}} = a\left(\frac{gH_s}{U_{10}^2}\right)^b \tag{16}$$

Buoy data show that the wave age of the data ranges from 0.5 to 100, there is a good correlation between the wave age and dimensionless wave height, and the final

fitting coefficient is $C_p/U_{10} = 3.37\left(gH_s/U_{10}^2\right)^{0.56}$. Combined with the method of Glazman and Greysukh [43], the equation can be described as (17), which is called the three-fifths index rate.

$$\frac{C_p}{U_{10}} = 3.31\left(\frac{gH_s}{U_{10}^2}\right)^{\frac{3}{5}} \tag{17}$$

Based on (16), Wang uses the equation, the peak wave period $T_p$ is converted into the $T_z$, and the parameters are finally set. Then, the fitting formula is obtained as follows [44]:

$$1.44\frac{gT_z}{2\pi U_{10}} = \alpha\left(\frac{gH_s}{U_{10}^2}\right)^\beta \tag{18}$$

The buoy data, except for in the Gulf of Mexico and Hawaii, are combined for fitting, and the fitting results are shown as follows:

$$1.44\frac{gT_z}{2\pi U_{10}} = 3.447\left(\frac{gH_s}{U_{10}^2}\right)^{0.623} \tag{19}$$

Among them, Toba's scheme uses friction velocity ($u_*$), so we use Weisberg's drag coefficient scheme of Equation (23) and $U_{10}$ to calculate the friction velocity ($u_*$). Other schemes will be used for fitting. In the fitted mean wave period elements, the wind speed and significant wave height data were used from the output data of the SWAN model. By comparing the data of eight KMA buoys, we obtain Figure 9.

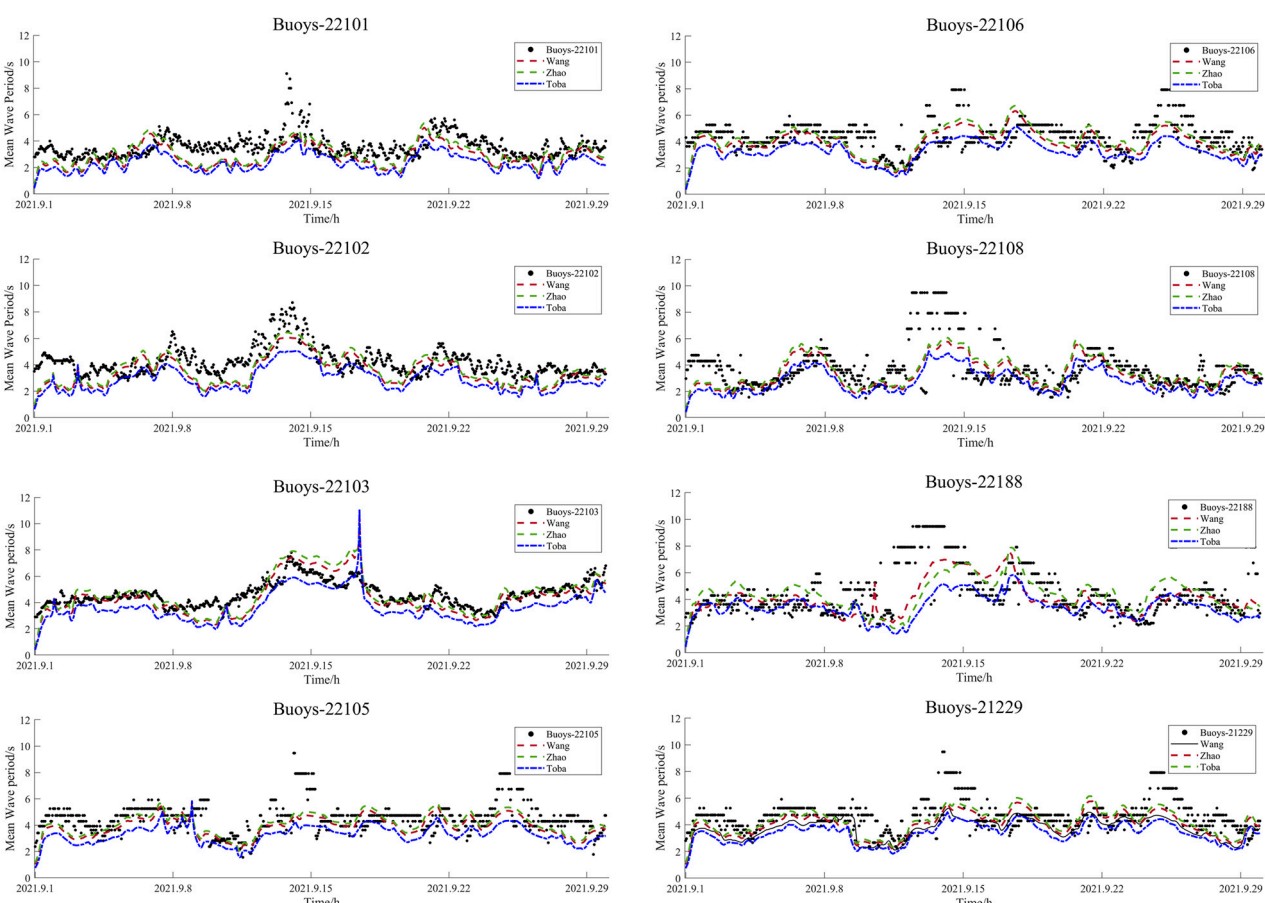

**Figure 9.** Previous parameterization and comparison with the SWAN model and KMA buoy data.

In Figure 9, the blue scatter points represent the KMA buoy observation data, the black solid lines represent the SWAN model output data, and the yellow dotted lines, green

dotted lines, and red solid lines represent the Toba, Zhao, and Wang parameterization schemes. The figure shows that the output values of the elements of the three curves are higher than those of SWAN, Toba's scheme is close to SWAN's mean wave period output, and Zhao's scheme has the highest overall average value, which is closest in the curve trend. Wang's fitting value is between the other two parameterization schemes. However, due to the characteristics of a single parameter, the relationship between the wind speed and significant wave height does not correspond when the components of the wave are rapidly converted; currently, abnormal fitting peaks appear in the curve. This phenomenon exists in all three schemes, so it is necessary to eliminate accidental errors by dividing fitting coefficients and distinguishing wave components.

*4.2. Parameter Application and Comparison*

By exploring the simulation of the wave field from the SWAN model, the mean wave period ($T_z$) of the typhoon wave field is generally lower than the measured value [13,33]. According to previous studies, there have been many parameterization schemes, such as the wave parameterization scheme and the three-second power law for wind waves of simple spectra proposed by Toba [42], which is the first generation of wave parameterization achievements. Zhao et al. used many buoy data to obtain a 3/5 exponential rate fit based on measured data [35]. Wang et al. used NDBC buoy data to simulate new parameterization data [44]. Wang et al. optimized Toba's scheme to obtain new nonpower exponential parameters based on a three-second power law [21]. Many of them are based on the relationship between the dimensionless wave height and wave age or dimensionless wave period. The relationship between them is shown as follows:

$$H^* = BT^{*A} \tag{20}$$

where $H^*$ is the dimensionless wave height and $T^*$ is the dimensionless wave period or wave age. *A* and *B* are the parameters from the scheme.

Furthermore, $H^* = gH_s/u_*^2$ and $T^* = gT_p/u_*$. $u_*$ is the friction velocity. Then, we use the correlation between $u_*$ and $u_{10}$. $T_p$ is the peak wave period, which represents the period corresponding to the maximum peak, where $T_p = 2\pi/\omega_p$, and $T_p$ can transform into $T_z$ according to Toba [45]:

$$T_p = 1.44\,T_z \tag{21}$$

(21) can transform into the following:

$$\frac{gH_s}{u_*} = B\left(\frac{gT_z}{u_*^2}\right)^A \tag{22}$$

According to the relationship, $u_*^2 = C_d \times u_{10}^2$, where $C_d$ is the drag coefficient, which was adopted by Weisberg and Zheng [46].

$$C_d \times 10^3 = \begin{cases} 1.2 & U_{10} < 11\ \text{m/s} \\ 0.49 + 0.065 \times U_{10} & 11\ \text{m/s} < U_{10} < 25\ \text{m/s} \\ 0.49 + 0.065 \times 25 & U_{10} > 25\ \text{m/s} \end{cases} \tag{23}$$

Combining the coefficients and replacing the coefficients in the formula with the $\alpha$ and $\beta$ parameters, we obtain the following fitting equation:

$$\frac{gT_z}{U_{10}} = \alpha\left(\frac{gH_s}{U_{10}^2}\right)^\beta \tag{24}$$

According to Hasselmann [41], the correlation between the dimensionless wave height and wave age is proposed as follows:

$$\frac{gH_s^2}{u_*^4} = 3.2 \times 10^{-3} \left(\frac{C_p}{u_*}\right)^3 \tag{25}$$

where $C_p$ is the phase velocity of significant waves, and $C_p = gT_p/2\pi$. It can be transformed as follows:

$$(H^*)^2 = g \times 3.2 \times 10^{-3}(T^*)^3 \tag{26}$$

According to the corresponding relationship and previous research, when the dimensionless wave height is less than 0.26, the wind wave acts significantly. When the wind wave begins to grow and the dimensionless wave height is greater than 0.26 but less than 1, the wind wave and swell interact as mixed waves. When the dimensionless wave height is greater than 1, the swell plays a significant role in the wave composition. We divide the wave composition into three cases and then establish a piecewise function to fit according to the correlation of the wave age. According to (26), when the dimensionless wave height is 0.26, the corresponding wave age is 1.29, and when the dimensionless wave height is 1, the corresponding wave age is 3.17. Finally, according to (24), through piecewise fitting, we obtain three sets of coefficients in the parameterization scheme.

After the parameterization process shown in these steps, we obtained three sets of parameters and tested the measured values according to the corresponding parameters. The following contents are different accuracy tests for different types of parameters. We used shallow-sea buoy data to test and compare the fitting coefficients of shallow-sea buoys. We use the typhoon observation data to test and compare the fitting coefficient between the deep sea and the total buoy data. The following are the details of the experiment.

Most of the nearshore buoy data we selected come from the eastern part of North America, where the latitude is between 30° and 40°. The data show that the wave direction has a concentrated distribution area of 100° and 150°, and the data source is shallow-sea buoys, which complies with the real situation of nearshore swells. Based on the characteristics of the wave age distribution, we use the data points in deep-sea buoys to fit the parameters of wind waves and use the data points in shallow-sea buoys to fit the parameters of swell waves. The other data points fit the parameters of mixed waves. To make the results more accurate, we set a restrictive condition, which distinguishes the wind direction and wave direction, and we classify the wave data with wave ages less than 1.29 and an angle between the wind direction and wave direction of less than 45 degrees as wind wave dominance; we classify the data with wave ages greater than 3.17 degrees as swell dominance. The remaining data are divided into mixed waves. This distinction makes the fitted parameters more precise. As shown in the data, most observed data are near the positive correlation curve of $k = 1$, which reflects the characteristics of wind waves. The shallow-sea buoy data are nearshore, and the wave direction of the swell data is mostly the same, which agrees with the actual nearshore wave propagation characteristics. To reflect the optimization of the SWAN model, all data sources come from the output of the SWAN model, including the mean wave period, significant wave height, and wind speed.

We selected 16 shallow-sea buoy data points as the fitting data for the swell and mixed wave parameters, with a total of 38,557 data nodes. After data processing, a total of 8769 effective computing nodes were obtained for fitting the swell parameters, and 16,555 effective computing nodes were obtained for fitting the mixed wave parameters. In addition, 12 deep-sea buoys were selected as the wind wave parameter fitting data, with a total of 42,614 data nodes. After data processing, a total of 12,069 effective data nodes were obtained. All selected data nodes have been linearly fitted, and the corresponding fitting coefficients are obtained as listed.

As Figure 10 shows, X is $ln\frac{gH_s}{U_{10}^2}$, Y is $ln\frac{gT_z}{U_{10}}$, the blue dots represent data nodes for fitting, and the red lines represent linear fitting lines. After fitting, the transformation relationship between the peak wave period and mean wave period is noted in (21).

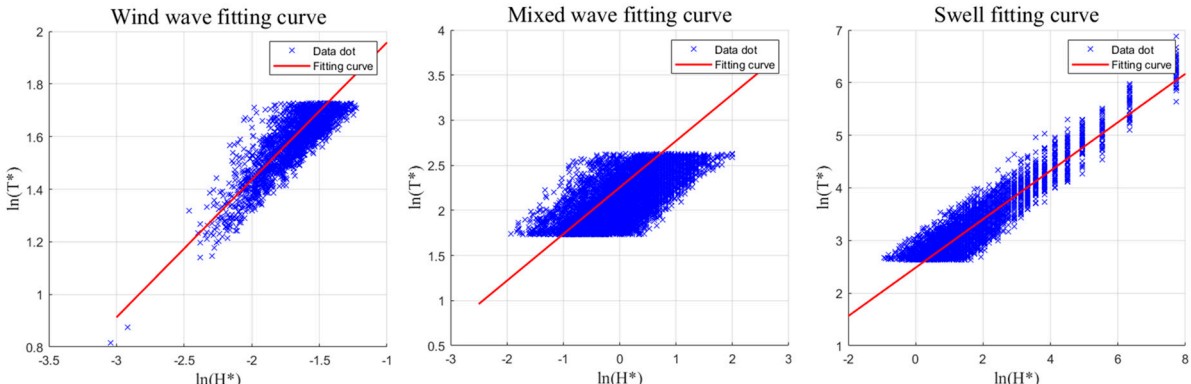

**Figure 10.** Fitting curve with three sets of parameters.

We change the whole fitting into a three-piece fitting according to the division of the wave age, and the obtained fitting parameters are shown in Table 5, in which the value of parameter $\alpha$ gradually decreases during the transition from wind wave to swell. We call this new scheme WS-23. The wave has a nonlinear effect during the transition from wind wave to swell, which leads to energy dissipation and slope reduction. The parameter $\beta$ shows little change, in which the value of $\beta$ has little change in the case of wind waves and swells and decreases in the case of mixed waves. Because Toba's exponential parameter is 0.666, while Zhao's and Wang's are lower than this value, the fitting result is worse than the other two schemes. Therefore, effective control of the $\alpha$ value can improve the fitting effect. According to previous research results, the parameter that determines the accuracy of fitting results is $\alpha$.

**Table 5.** Three types of parameters with linear fitting.

| Parameter | $\alpha$ | $\beta$ | $SE\ (\alpha)$ | $SE\ (\beta)$ |
|---|---|---|---|---|
| Swell | 12.982 | 0.5170 | 0.0035 | 0.0064 |
| Mixed wave | 13.611 | 0.5227 | 0.0073 | 0.0081 |
| Wind wave | 14.585 | 0.5747 | 0.0070 | 0.0104 |

In Table 5, there are also two sets of *SE* (standard error) for the data values, which correspond to the estimated standard errors of different parameters, and this value measures the average deviation between the observed values distributed around the sample regression line and the fitted values on the line. The data show that the alpha coefficient, except for the wind and waves, is 0.0170, and the *SE* values of the other coefficients are lower than 0.01. The fitting of the swell coefficient is the best, and the lowest value reaches 0.0018, which shows that the fitting result is good.

Next, we use Wang and Zhao's scheme for reference, test the new parameterized scheme, WS-23, and participate in the data comparison of the SWAN model data. Figure 11 shows the comparison results of the data. The red line represents the fitting data of the WS-23 scheme, and the other curves represent the data or other schemes. From the curve trend, WS-23 has the same trend as the other schemes; however, due to the limitation of the wave age, the abnormal peak brought by fitting is eliminated. In most cases, the fitting values of the WS-23 scheme are larger than those of the other schemes, and at the same time, they are more consistent with the buoy observation data.

Table 6 presents the error analysis data obtained from different schemes. The data clearly demonstrate that, after comparing the buoy data of WS-23 with other parameterization schemes, the new parameter results for all buoy data are significantly better than those of other schemes. Additionally, the fitting value *r* for all new buoy data schemes has improved, with the largest improvement for Buoy-22108, where the r increased by 0.11. The $E_a$ value decreased by 0.26, the $E_c$ value decreased by 0.07, and the *RMSE* value decreased by 0.23, enhancing the correlation between data points. Compared to other schemes, the mean

$E_a$, $E_c$, $r$, and $RMSE$ values for the remaining buoys have been optimized. Specifically, relative to the SWAN mode output value, the mean values of $E_a$, $E_c$, and $RMSE$ decreased by 0.231, 1.94%, and 0.162, respectively, while $r$ increased by an average of 0.05. The data and correlation analysis confirm that the WS-23 scheme is feasible and demonstrate its strong ability to optimize the SWAN model's output compared to other schemes.

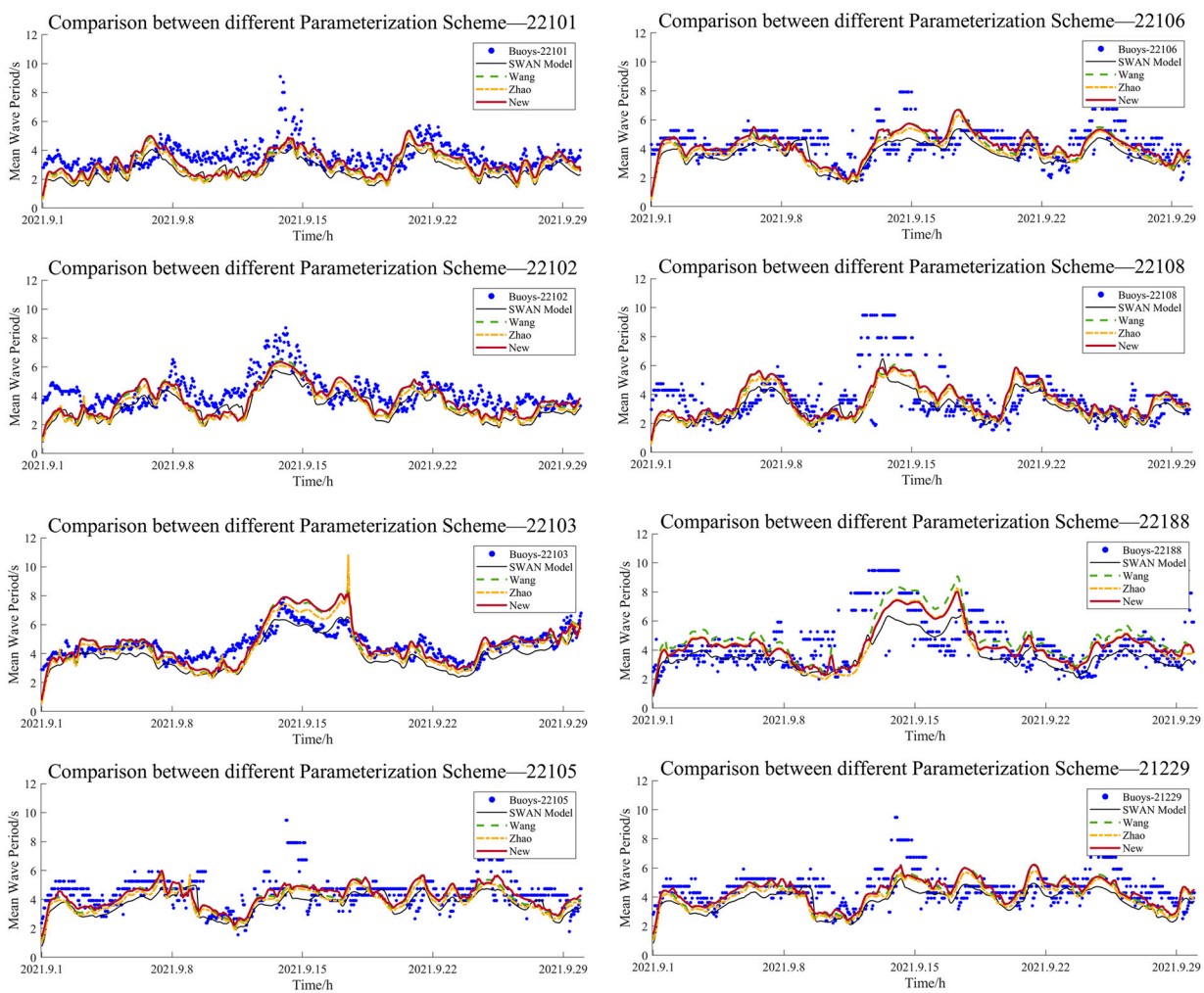

**Figure 11.** Mean wave period comparison between the SWAN model output, WS-23 scheme, and other parameterization schemes from previous results.

**Table 6.** Error analysis with different types of parameters.

| Buoys | $E_a$ | | | | $E_c$ | | | | $RMSE$ | | | | $r$ | | | |
|---|---|---|---|---|---|---|---|---|---|---|---|---|---|---|---|---|
| | SWAN | Zhao | Wang | WS-23 | SWAN | Zhao | Wang | WS-23 | SWAN | Zhao | Wang | WS-23 | SWAN | Zhao | Wang | WS-23 |
| 22101 | 1.04 | 0.84 | 0.92 | 0.78 | 0.29 | 0.23 | 0.25 | 0.22 | 1.21 | 1.03 | 1.11 | 0.97 | 0.47 | 0.48 | 0.49 | 0.56 |
| 22102 | 1.04 | 0.98 | 1.04 | 0.92 | 0.24 | 0.23 | 0.25 | 0.22 | 1.25 | 1.15 | 1.22 | 1.09 | 0.59 | 0.61 | 0.61 | 0.67 |
| 22103 | 0.75 | 0.69 | 0.67 | 0.65 | 0.17 | 0.15 | 0.15 | 0.14 | 0.87 | 0.88 | 0.84 | 0.83 | 0.84 | 0.83 | 0.83 | 0.86 |
| 22105 | 0.98 | 0.87 | 0.94 | 0.83 | 0.20 | 0.19 | 0.20 | 0.18 | 1.31 | 1.13 | 1.21 | 1.09 | 0.48 | 0.49 | 0.49 | 0.51 |
| 22106 | 0.95 | 0.83 | 0.88 | 0.79 | 0.20 | 0.19 | 0.20 | 0.18 | 1.21 | 1.07 | 1.12 | 1.03 | 0.54 | 0.53 | 0.54 | 0.57 |
| 22108 | 1.23 | 1.34 | 1.27 | 1.20 | 0.23 | 0.31 | 0.27 | 0.26 | 1.74 | 1.73 | 1.73 | 1.62 | 0.52 | 0.50 | 0.47 | 0.58 |
| 22188 | 1.09 | 1.07 | 1.08 | 1.05 | 0.26 | 0.28 | 0.27 | 0.28 | 1.56 | 1.50 | 1.52 | 1.47 | 0.52 | 0.52 | 0.53 | 0.58 |
| 21229 | 1.03 | 0.85 | 0.92 | 0.82 | 0.21 | 0.18 | 0.19 | 0.17 | 1.32 | 1.09 | 1.17 | 1.06 | 0.52 | 0.53 | 0.54 | 0.54 |

## 5. Conclusions

We use ERA5 wind data from the ECMWF as model data to drive and test the operation effect of the model and the stability of the unstructured grid using wave parameter data (NMDC) from nearshore stations and significant wave height data from a satellite altimeter (NCEI, Jason-3). The wind speed is compared and tested. After inspection, the output

elements of the model are normal, and the stability of the unstructured grid is good. From the test results, the simulation effect of the typhoon wave is good, the identification ability of the wind circle and element gradient field is strong, and the test effect of the simulated significant wave height of the element is good for the measured data; however, the simulation effect of the mean wave period is poor.

Then, we reproduce the mean wave period parameterization schemes fitted by predecessors and fit the schemes of Toba, Zhao, and Wang. The output results of Zhao and Wang's parameterization schemes for SWAN are better; however, some curves of the fitted mean wave period elements have abnormal peak characteristics, and the parameterization parameters are solitary, which cannot distinguish wind waves from swells, so the nearshore buoys and offshore buoys are fitted together without considering the composition of wave components. However, considering the generation and conversion mechanism of wind waves and swells, we introduce the concept of wave age and distinguish wind waves, swells, and mixed waves based on differences in wave age, wind direction, and wave direction to improve the optimization degree and accuracy of the new parameterization scheme in the nearshore area. Based on the buoy data (NDBC), the coefficients are fitted. Compared with other schemes, the mean wave period elements fitted with the new parameterized scheme are improved in terms of the mechanism, and the error analysis data are all improved, which shows that they are more consistent with the measured values and the trend is more consistent.

The innovation of this research lies in the use of a wave age relationship to differentiate between wind waves and swells. We utilize SWAN output data and parameterization relationships to calculate wave age, and then, we distinguish the parameters to determine the $T_z$. All the data used in the calculation process originate from SWAN, such as wind speed, $H_s$, and $T_z$. To calculate the wave age, we require the mean wave period output of SWAN to calculate the mean wave period of the new scheme. To enhance the reliability of the model output, iteration can be performed. The higher the number of iterations is, the more accurate the calculated results become. The novel parameterization scheme proposed in this paper has distinct characteristics between wind waves and swells, making it suitable for sea areas with complex wave compositions and nearshore fitting.

**Author Contributions:** Conceptualization, J.L. and W.Z.; formal analysis, J.L. and H.W.; methodology, W.Z. and J.W.; project administration, J.S.; resources, W.Z., J.S. and J.W.; data curation, J.L. and X.C.; validation, J.L.; visualization, J.L., H.W. and Q.W.; writing—original draft preparation, J.L.; writing—review and editing, J.L., W.Z., J.S., J.W., H.W., X.C., Q.W. and Z.Z.; supervision, W.Z. and J.S. All authors have read and agreed to the published version of the manuscript.

**Funding:** This research was funded by the National Science Foundation of China (Grant No. 42192552).

**Data Availability Statement:** The typhoon track data come from the CMA Tropical Cyclone Data Centre (https://tcdata.typhoon.org.cn/, accessed on 1 July 2023). The fitted buoy data are selected from the National Data Buoy Centre (NDBC) (https://www.ndbc.noaa.gov/, accessed on 15 July 2023). The comparison buoy data come from the Korea Meteorological Administration (KMA) (https://data.kma.go.kr/data/sea/, accessed on 15 July 2023). The SWAN model offshore inspection data are taken from the National Marine Data Centre (NMDC) (https://mds.nmdis.org.cn/, accessed on 1 July 2023), and the offshore detection data come from the National Centres for Environmental Information (NCEI) (https://www.ncei.noaa.gov/data/oceans/jason3/gdr_f/gdr_ssha/, accessed on 15 July 2023). The wind field data are provided by the European Centre for Medium-Range Weather Forecasts (ECMWF) (https://cds.climate.copernicus.eu/cdsapp#!/home, accessed on 1 July 2023). Coastline data were taken from the Global Self-consistent, Hierarchical, High-resolution Geography Database (GSHHG) (https://www.soest.hawaii.edu/pwessel/gshhg/, accessed on 1 July 2023).

**Acknowledgments:** The authors thank the National Weather Service for providing the typhoon traverse route, the reanalysis wind field data provided by the ECMWF, the station observation data provided by the NCEI, the buoy observation data provided by the KMA and NDBC, the altimeter data provided by the NCEP, and the coastline data provided by the GSHHG.

**Conflicts of Interest:** The authors declare no conflict of interest.

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
