# Peer review of "The Wave Period Parameterization of Ocean Waves and Its Application to Ocean Wave Simulations"

_remotesensing, doi:10.3390/rs15225279_

Round 1
Reviewer 1 Report
Comments and Suggestions for Authors
Comments:
This paper presented an investigation on wave period parameterization based on wave numerical simulation, nearshore observation, and buoys as well as satellite altimeter data. Compared to the traditional wind wave parameterization, the corresponding research on swell and mixed wave is also taken into account. The comparison of the correlation coefficient between simulation and experimental results validates the new parameterization scheme. The paper is well organized and presented, which can be recommended for publication.
Author Response
Dear Editor and Reviewers:
On behalf of my co-authors, we thank you very much for giving us an opportunity to revise our manuscript, and we also appreciate reviewers very much for their positive and constructive comments and suggestions on our manuscript entitled “Wave period parameterization of ocean waves and application in ocean wave simulation”. (Manuscript Number: 2602259).
We revised the manuscript according to these comments and suggestions. In general, we have tried our best to revise our manuscript and provide the point-by-point responses. All changes were marked in red. Attached please find our responses to the referees’ comments.
The following is a summary of main changes:
- We have rewritten Section 1 - Introduction and Section 2 - Materials and Methods, making some corrections in the manuscript. The revision includes the removal of irrelevant content from Section 1 and the addition of an introduction to the original function terms of the SWAN model.
- We have revised the parameterized content in Section 4 - Discussion, including redrawing images (such as Figure 9), accurately expressing parameters, explaining original data and adding summary.
- We have moved the paragraph related to the parameterization scheme to Section 4.2 and made modifications to the topic of parameterization.
- We made correction to Abstract and emphasize the highlight of the manuscript.
- We standardize the expression of all parameters, including the expression of ocean wave elements and error analysis parameters (such as relative error and correlation coefficient). And we checked all the typos or grammar mistakes and polished the whole paper and fixed language errors.
- Incorrect data has been deleted, specifically Buoy-22107 data, and replaced with new data from Buoy-22188. After conducting model simulations, we have generated additional data and replotted certain figures and tables, including Figure 9, Figure 11 and Table 5.
- We have added the Section: Author Contribution and Funding.
Once again, thank you very much for your comments and suggestions. And we hope that the revised manuscript can be accepted by Special Issue “Advances in the Ocean Surface Dynamics: Ocean Waves, Wind, and Air-Sea Interaction - in Memory of Professor Shengchang Wen.”
Thank you and best regards!
Yours sincerely, Jialei Lv.
E-mail: lv15344179880@163.com
Corresponding author:
Name: Wenjing Zhang
E-mail: zhangwenjing21@nudt.edu.cn

Reviewer 2 Report
Comments and Suggestions for Authors
The originality of this paper lies in the introduction of a new parameterisation scheme (WS-23) into the SWAN model in order to obtain a better simulation of the mean wave period. The new parameterisation introduces the concept of wave age and distinguishes between wind waves, swell waves and mixed waves in order to improve the degree of optimisation and the accuracy of the new parameterisation scheme in the near shore area. The rest of the paper follows the steps proposed by other papers, as explained by the authors. . The authors state that "compared to other schemes, the wave period elements fitted by the new parameterised scheme are improved in mechanism and the correlation coefficients are all improved, which shows that they are more consistent with the measured values and the trend is more consistent". This consideration is too strong because, using Wang and Zhao's scheme as a reference, the results of the new parameterisation of WS-23 don't show a relevant improvement, but the results seem very similar to the reference one.
Comments on the Quality of English LanguageThe main problem is the lack of precision with which the paper is written. The quality and style of the language is poor, making it difficult to understand the aim and research plan. Sentences are sometimes very unclear and there are typing errors in formulae and text.
Author Response
Dear Editor and Reviewers:
On behalf of my co-authors, we thank you very much for giving us an opportunity to revise our manuscript, and we also appreciate reviewers very much for their positive and constructive comments and suggestions on our manuscript entitled “Wave period parameterization of ocean waves and application in ocean wave simulation”. (Manuscript Number: 2602259).
We revised the manuscript according to these comments and suggestions. In general, we have tried our best to revise our manuscript and provide the point-by-point responses. All changes were marked in red. Attached please find our responses to the referees’ comments.
The following is a summary of main changes:
- We have rewritten Section 1 - Introduction and Section 2 - Materials and Methods, making some corrections in the manuscript. The revision includes the removal of irrelevant content from Section 1 and the addition of an introduction to the original function terms of the SWAN model.
- We have revised the parameterized content in Section 4 - Discussion, including redrawing images (such as Figure 9), accurately expressing parameters, explaining original data and adding summary.
- We have moved the paragraph related to the parameterization scheme to Section 4.2 and made modifications to the topic of parameterization.
- We made correction to Abstract and emphasize the highlight of the manuscript.
- We standardize the expression of all parameters, including the expression of ocean wave elements (such as ) and error analysis parameters (such as relative error () and correlation coefficient ()). And we checked all the typos or grammar mistakes and polished the whole paper and fixed language errors.
- Incorrect data has been deleted, specifically Buoy-22107 data, and replaced with new data from Buoy-22188. After conducting model simulations, we have generated additional data and replotted certain figures and tables, including Figure 9, Figure 11 and Table 5.
- We have added the Section: Author Contribution and Funding.
Once again, thank you very much for your comments and suggestions. And we hope that the revised manuscript can be accepted by Special Issue “Advances in the Ocean Surface Dynamics: Ocean Waves, Wind, and Air-Sea Interaction - in Memory of Professor Shengchang Wen.”
Thank you and best regards!
Yours sincerely, Jialei Lv.
E-mail: lv15344179880@163.com
Corresponding author:
Name: Wenjing Zhang
E-mail: zhangwenjing21@nudt.edu.cn

Reviewer 3 Report
Comments and Suggestions for Authors
Please find the attachment.

The proposed wave period parameterization is interesting and practicable, but the writing is really poor.
Author Response

(The authors gave the same response as above.)

Reviewer 4 Report
Comments and Suggestions for Authors
The paper develops a new parameterization scheme of surface wave period. In the primary evaluation, the new parameterization scheme outperforms third-generation wave model and other parameterization schemes. The results are valuable for surface wave study.
[1] Caption of Table 1. “traverse route” should be “best-track”.
[2] Figure 5. Y coordinate is SWAN output ? Or ERA reanalysis?
[3] Figure 9 and Figure 11 is similar. Probably Figure 9 is not necessary.
[4] Table 5 shows the new parameterization scheme? How to use new parameterization scheme in prediction? Please explain this.
[5] Language should be further improved.
Comments on the Quality of English LanguageLanguage should be further improved.
Author Response

(The authors gave the same response as above.)

Round 2
Reviewer 2 Report
Comments and Suggestions for Authors
Attached you'll find detailed comments and suggestions for authors
Regards

Author Response
Dear Editor and Reviewer:
On behalf of my co-authors, we thank you very much for giving us an opportunity to revise our manuscript, and we also appreciate reviewer very much for those positive and constructive comments and suggestions on our manuscript entitled “Wave period parameterization of ocean waves and application to ocean wave simulations”. (Manuscript Number: 2602259).
We revised the manuscript according to these comments and suggestions. In general, we have tried our best to revise our manuscript and provide the point-by-point responses. All changes were marked in red. Attached please find our responses to the referees’ comments.
The following is a summary of main changes:
- We have checked the grammar in the manuscript and corrected the writing and citation errors that pointed out by the Reviewer.
- We have thoroughly examined all the wave parameters, including all wave height and wave period parameters. We have provided explanations for all these wave parameters and used them consistently throughout the manuscript.
- We have made revisions to some of some Figures and Tables as needed in the manuscript.
Once again, thank you very much for your comments and suggestions. And we hope that the revised manuscript can be accepted by Special Issue “Advances in the Ocean Surface Dynamics: Ocean Waves, Wind, and Air-Sea Interaction - in Memory of Professor Shengchang Wen.”
Thank you with best regards and look forward to your reply!
Yours sincerely, Jialei Lv.
E-mail: lvjialei18@nudt.edu.cn

Reviewer 3 Report
Comments and Suggestions for Authors
I am satisfied with the response to my comments.
Comments on the Quality of English LanguageMinor editing of English language is required.